# Oral Health Status and the Impact on Oral Health-Related Quality of Life among the Institutionalized Elderly Population: A Cross-Sectional Study in an Area of Southern Italy

**DOI:** 10.3390/ijerph18042175

**Published:** 2021-02-23

**Authors:** Aida Bianco, Silvia Mazzea, Leonzio Fortunato, Amerigo Giudice, Rosa Papadopoli, Carmelo Giuseppe Angelo Nobile, Maria Pavia

**Affiliations:** 1Department of Health Sciences, School of Medicine, University of Catanzaro “Magna Græcia”, Viale Europa, 88100 Catanzaro, Italy; a.bianco@unicz.it (A.B.); mazzea.silvia@studenti.unicz.it (S.M.); rosa.papadopoli@studenti.unicz.it (R.P.); 2Department of Health Sciences, School of Dentistry, University of Catanzaro “Magna Græcia”, Via T. Campanella, 88100 Catanzaro, Italy; leo@unicz.it (L.F.); a.giudice@unicz.it (A.G.); 3Department of Pharmacy, Health and Nutritional Sciences, University of Calabria, 87100 Cosenza, Italy; carmelo.nobile@unical.it; 4Department of Experimental Medicine, University of Campania “Luigi Vanvitelli”, Via L. Armanni, 5, 80138 Naples, Italy

**Keywords:** elderly, GOHAI, oral health, oral health impact, quality of life

## Abstract

**Background:** The objectives of this study were to describe the oral health status in the institutionalized geriatric population in an area of southern Italy and to identify the impact of oral health on the Oral Health Related Quality of Life (OHRQoL). **Methods:** Data were collected from individuals aged ≥60 years in randomly selected Calabrian long-term care facilities. The dental health status was assessed recording the decayed, missing, or filled dental elements due to the carious lesions (DMFT) index, the presence of visible dental plaque, and the gingival condition. The influence of the dental health status on the self-perceived value of life was assessed using the Geriatric Oral Health Assessment Index (GOHAI). **Results:** Among the 344 elderly individuals included, 18.4% reported frequent tooth-brushing, and only 39.9% reported the need of dental care. The DMFT index was 26.4. Less than a third of the participants had a GOHAI score of ≤50 which is suggestive of highly compromised OHRQoL. The GOHAI score was significantly better for elderly individuals with no self-perceived need of dental care and with a lower DMFT index. **Conclusions:** The burden of oral conditions among residents in long-term care facilities was considerable, with a high prevalence of missing teeth and dentures. Strategies targeting care providers are needed.

## 1. Introduction

In the last decades, life expectancy in developed countries has sharply risen and the proportion of people over 60 years of age within the population is increasing [1]. Italy is the “second oldest country in the world”, and life expectancy at 65 years is one year longer than the mean European Union value [2].

The health problems arising as a result of the aging process require special attention, considering that as people get older, they suffer from several chronic diseases that could influence, among other things, the ability to maintain oral hygiene; furthermore, the resulting use of medications may also endanger oral health [3,4]. This is of importance since maintenance of oral health and attention to dental care improve the function of the stomatognathic apparatus, interpersonal relationships, and therefore the overall quality of the subject’s life [5]. 

Oral Health Related Quality of Life (OHRQoL) is defined as an individual´s assessment of how functional, psychological and social factors, and experience of pain/discomfort in relation to orofacial concerns affect the well-being of that individual [6]. Thus, OHRQoL is a significant predictor of general health, emphasizing the need to maintain good oral health particularly in old age [7,8]. This is especially crucial for medically compromised or institutionalized elderly patients because the impact on them appears to be more severe. The long-term facility is a setting where subjects depend on help with daily living activities and/or need some permanent nursing care. Especially individuals over 75 years of age are more likely to develop chronic pathologies, comorbidities, or other impairing diseases that require continuous assistance until death. Moreover, there is a profound gap in the access to dental treatments between free living and institutionalized elders [9,10]. The latter have shown worse oral health due to their condition of fragility and to the environment in which they live where oral hygiene and dental care are not considered as a priority [11,12]. Several oral health status indicators have been developed [13,14,15,16,17], and the Geriatric Oral Health Assessment Index (GOHAI) was validated for the self-assessment of the oral status in elderly individuals [13].

The objectives of this study were to describe the oral health status evaluated by means of an oral examination, and to identify the impact of oral health on the OHRQoL, as assessed by the GOHAI, among the institutionalized elderly population in an area of southern Italy.

## 2. Methods

### 2.1. Study Population and Sampling

The survey was conducted during the period July 2018–June 2019. To overcome the difficulty of identifying a sampling frame of all institutionalized elderly individuals in Calabrian long-term care facilities, the whole region was divided into two areas (North and South), and in each area, a random sample of ten long-term care facilities (clusters) was selected from a publicly available frame of all facilities accredited by the Regional Health System. Then, all individuals within these clusters were listed, and data were collected from every institutionalized elderly individual aged ≥60 years, who was able to give written informed consent. Persons who were uncooperative and who could not understand or had a poor understanding of the Italian language were excluded. The sample size was determined in order to warrant estimation with an expected margin of error of 5%, assuming an intended confidence level (CI) of 95%. The prevalence of individuals who had a poor GOHAI score (25%) obtained from a similar study [18] was used. Based on these assumptions, a sample of at least 288 institutionalized elderly individuals was required. The cluster structure of the data was taken into account when calculating the sample size, whilst the magnitude of the design effect due to cluster sampling was estimated to be low and set at 1.2 [19]. This choice was driven by the consideration that the intracluster correlation could be considered low, since no substantial differences in the characteristics of the elderly institutionalized in the different facilities were expected. Given these assumptions, the sample size calculation yielded the need of 345 subjects. 

### 2.2. Data Collection

Data were collected from questionnaires administered by four trained and calibrated interviewers and from a clinical oral examination performed at the long-term care facility. The training involved presentation and explanation of the instrument and a practical phase (practice with other interviewers, shadow interview, reverse shadow). To assure the calibration, each interviewer repeated 10 interviews after one week in order to analyze the intra- and inter-rater agreement. Elderly subjects who had moderate or severe cognitive impairment were assisted by their caregivers in answering the questions about oral hygiene habits and GOHAI items, for preventing information bias.

### 2.3. Survey Instrument

Participants were asked to respond to questions relating to their socio-demographic status (age, gender, marital status, level of education, residential area, previous occupation), smoking habits and health status (chronic diseases), dental attendance and oral hygiene behavior (tooth brushing frequency, use of mouthwash and tongue-scraping habit). Eventual cognitive impairments evaluated by clinicians using tests of cognitive function, such as the Mini-Mental State Exam (MMSE) [20] or the Short Portable Mental Status Questionnaire (SPMSQ) [21] was retrieved from the patient’s medical records. A score of MMSE ≥24 suggests no cognitive impairment, while a score of 19 to 23 suggests mild cognitive impairment, a score of 10 to 18 suggests moderate cognitive impairment, and a score of ≤9 indicates severe cognitive impairment; regarding SPMSQ, a score of ≤2 indicates no cognitive impairment, a score of 3 to 4 mild cognitive impairment, and scores of 5 to 7 and ≥8 moderate or severe cognitive impairment, respectively. The daily frequency of tooth brushing was investigated with a scale of values expressed as never, not every day, once/day, 2 times/day, 3 or more times/day. Participants were also questioned about their feelings about their need for dental treatment. The GOHAI was used for the assessment of perceived oral health status [13], and consisted of 12 items reflecting problems that had affected the elderly individuals in the past 3 months, relating to three domains which are physical function, including eating, speaking, and swallowing; psychosocial function, including worry or concern about oral health, self-image, self-consciousness about oral health, and avoidance of social contacts because of oral problems; and pain or discomfort with regard to dental conditions on a 5-point scoring scale (1 = always, 2 = often, 3 = sometimes, 4 = rarely, and 5 = never). Questions about swallowing food, eating without discomfort, and about satisfaction of one’s own mouth are reverse-scored (1 = never and 5 = always). The GOHAI score was calculated by summing the scores to each statement, and it ranged from 12 to 60. It was divided into three categories: good (57–60), fair (51–56), and poor (less than 50) rating of oral health, indicating, respectively, good, moderate, and highly compromised OHRQoL [13]. 

### 2.4. Psychometric Properties of GOHAI

Preliminary analysis of psychometric properties were conducted. The reliability of GOHAI was assessed in terms of internal consistency, item-scale correlation, and test–retest coefficients. Internal consistency was achieved through the evaluation of Cronbach’s alpha among the twelve items of GOHAI. Item-scale correlation was measured by Pearson’s correlation coefficient to assess correlation between the items and the total score. Test–retest reliability was assessed by intraclass correlation coefficient (ICC) through an additional interview of 50 institutionalized elderly individuals at an interval of three weeks from the time of the first administration to test stability of the index over time. Participants in the pretesting phase were not considered as a part of the main study. Validity was investigated in terms of cross-cultural and construct validity, considered in the subcategories of convergent, discriminant, and factorial validity. Cross-cultural validity was applied as per the procedure of forward backward translation and adaptation protocol to the Italian culture recommended by the World Health Organization [22]. Convergent validity of the GOHAI was examined by computing Pearson’s correlations among the GOHAI score and DMFT, DT, MT, FT, GI, PI, since it was expected that lower GOHAI scores would be associated with poor oral health objectively measured by the dentist. Discriminant validity was assessed by comparing the GOHAI score and self-reported oral hygiene habits which should not be notably associated with OHRQoL. The factorial validity was verified by using the principal-component factor method to analyze the correlation matrix. Furthermore, a varymax rotation was applied to maximize the separation of variables concerning the single factors.

### 2.5. Oral Examination

The oral examinations were performed in medical examination rooms provided by the institution that hosted the patients the same day as the questionnaire was administered using pre-packaged sterilized portable dental equipment. The clinical evaluation of patients was performed by an expert dentist through a thorough examination of the oral cavity for the evaluation of the oral status. The decayed, missing, and filled teeth (DMFT) and surfaces (DMFS) indices were used to record caries prevalence (or experience) [23]. The former refers to the sum of decayed, missing, or filled dental elements due to carious lesions. The basis for DMFT calculation is 32 teeth, i.e., all permanent teeth including wisdom teeth. High scores indicate worse dental health. The DMFS index expresses the number of affected tooth surface. The oral examination was performed in accordance with the WHO standardized methodology [23]. Dental practitioner also recorded whether the patient was edentulous or wore dentures. Dental plaque and gingival condition was also evaluated using previously published criteria [24,25]. The following scoring system for the Gingival Index (GI) was used: 0 = normal gingiva; 1 = mild inflammation (i.e., slight change in color, slight edema, no bleeding on probing); 2 = moderate inflammation (i.e., redness, edema, and glazing, or bleeding on probing); 3 = severe inflammation (i.e., marked redness and edema, tendency toward spontaneous bleeding, ulceration). The Plaque Index (PI) uses the following scoring system: 0 = absence of microbial plaque; 1 = thin film of microbial plaque along the free gingival margin; 2 = moderate accumulation with plaque in the sulcus; 3 = large amount of plaque in sulcus or pocket along the free gingiva margin. Similarly to the GI, the PI is scored on four sites per tooth of six index teeth. If any of these six teeth were missing, the opposing tooth was assessed, or if it was also missing one of the adjacent teeth was examined [26]. GI and PI were not evaluated in edentulous elderly individuals.

Approval from the Regional Ethics Committee was obtained (ID No. 44/2018/2/22).

### 2.6. Data Analysis

A stepwise multiple logistic regression model was carried out to assess the independent effect of several covariates after adjusting for the effect of confounders on the following outcome: overall impact score measured with GOHAI in the past 3 months (0 ≤ 50, 1 > 50). The following predictor variables were included in the model: age (continuous), gender (1 = male, 2 = female), marital status (1 = widower, 2 = others), education level (nominal: 1 = none, 2 = primary/middle school, 3 = high school/university degree), previous occupation (1 = employed, 2 = unemployed and housewife), smoking habit (1 = never smoker, 2 = past or current smoker), self-reported need of dental care (1 = no, 2 = yes), cognitive impairment (1 = severe cognitive impairment, 2 = moderate cognitive impairment, 3 = mild cognitive impairment, 4 = no cognitive impairment), use of prosthesis (nominal: 1 = no prosthesis, 2 = removable partial denture, 3 = removable complete denture, 4 = fixed partial denture), DMFT (continuous), GI [0 = normal gingival, 1 = mild inflammation (i.e., slight change in color, slight edema, no bleeding on probing), 2 = moderate inflammation (i.e., redness, edema, and glazing, or bleeding on probing), 3 = severe inflammation], and PI (0 = absence of microbial plaque, 1 = thin film of microbial plaque along the free gingival margin, 2 = moderate accumulation with plaque in the sulcus, 3 = large amount of plaque in sulcus or pocket along the free gingiva margin). Explanatory variables that were not related to GOHAI scores in the univariate analysis with *p*-values >0.25 were not included in the logistic regression. The significance level for variables entering the logistic regression model was set at 0.2 and for removing from the model at 0.4. The *kappa* coefficient to calculate the intra- and inter-rater agreement was used.

Data were stored and analyzed using Stata Statistical Software Version 14.1 (Texas, USA) [27]. The dataset was deposited in Mendeley Data repository (doi:10.17632/2ww38jd2db.1).

## 3. Results

### 3.1. Patient Characteristics

After screening for the exclusion criteria, 344 individuals were eligible and were invited to participate in the study; of 344 elders invited to participate in the study, 288 answered the general questionnaire and underwent oral examination (84.5% response rate). The reported reasons for not participating in the study were “for health reasons”, “no interest”, and “no specific reasons”. The mean age of the participants was 82.7 years (±9.9, range 61–99 years) and the majority were in the ≥80 years age group (70.5%); there were 195 (67.7%) women and 93 (32.3%) males. Nearly three quarters of them had no or primary level of education, and 51.7% were widowers. Seventy-one percent of the participants had never smoked, 41.7% and 23.3% had neurological or cardiovascular disease, respectively. The MMSE or SPMSQ were completed by 264 (91.7%) elderly subjects. Fifty-two elderly subjects (19%) had no cognitive impairment, whereas 52.2% had moderate or severe cognitive impairment. When looking at oral hygiene habits, 18.4% of the participants reported frequent tooth-brushing. Among those who wore a denture, 90.1% reported cleaning dentures once a day. No differences in the proportion of elders with low GOHAI score were observed between categories of tooth-brushing. Only 39.9% participants reported the need of dental care (Table 1). 

### 3.2. Oral Health Conditions and GOHAI Distribution

The intra- and inter-rater agreement rates varied from 0.891 to 0.948. Overall, less than a third (29.2%) of the participants had a GOHAI score of ≤50 (average 53.3 ± 6.3 and median 56) that is suggestive of highly compromised OHRQoL.

Table 2 shows the distribution of GOHAI by physical function, pain or discomfort, and psychosocial function. Elderly individuals with the lowest GOHAI mean scores were having difficulty in eating certain foods (3.6 ± 1.3) and could not eat certain foods or the desired amount of them (3.7 ± 1.3). The majority of the sample (98.6%) found food easy to swallow. Only 13.9% self-reported problems with verbal communication that prevent them from speaking as clearly as they wished. 

Regarding the pain and discomfort dimension, only 6.6% had pain or discomfort while eating, 5.2% of the participants declared they took medication at least sometimes to relieve feelings of pain or discomfort in their mouth, and 14.9% had sensitivity to hot, cold, or sweet food.

Regarding the GOHAI score associated with psychosocial function, almost one third (32.3%) were worried about the problems with their mouth and teeth, and 18% of the sample felt uncomfortable or stressed because of the condition of their mouth and teeth. A large proportion of subjects (95.2%) were found to be never or rarely embarrassed to eat in front of others because of the condition of their mouth and teeth, and 89.2% were satisfied with the condition of their mouth and teeth. A total of 31.9% of participants were edentulous and among them 26.1% did not wear any dentures, whereas in the overall sample, 27.1% and 6.9% wore a complete and partially removable denture, respectively, and 8% wore a fixed partial denture. Among those who had not prosthesis (167), only one subject had all natural teeth. The mean DMFT index was 26.4 (±7.5), and the mean decayed, missing, and filled teeth index values were respectively 3.5 (±4.6), 22.5 (±9.5), and 0.3 (±1.4). The analysis of the distribution of the various components of the DMFT score showed that missing teeth was the most prevalent characteristic with all subjects that had one or more teeth extracted due to caries. A total of 239 participants (83%) had more than 12 missing teeth. Only 50.4% had at least one filled tooth and 70.8% had one or more decayed teeth. Twenty-two point four percent of the participants were free of visible dental plaque on any of the index teeth and 87 subjects (41.4%) had a PI value of 2. The mean PI was 2.23. Only 2.4% of subjects showed that they had a healthy gingival condition and five elders reported the higher GI value. The average of GI was 1.77.

Table 3 presents the results of the multivariate logistic regression analysis, which indicated that a high GOHAI score was significantly associated with no self-perceived need of dental care (OR = 0.20, 95% CI = 0.97–0.42). Moreover, the GOHAI score was significantly higher for elderly individuals with a lower DMFT index; hence, each unit increase in the DMFT index causes a 12% decrease in the odds of a good GOHAI score (OR = 0.88, 95% CI = 0.83–0.94). Finally, smoking and other socio-demographic variables were not associated to GOHAI. 

### 3.3. Psychometric Properties of GOHAI

The internal consistency of the GOHAI domains and total score were high (Cronbach’s α coefficients ranged from 0.78 to 0.85 for subscales and 0.93 for total score) and test–retest reliability was acceptable (in the range of 0.71–0.92). Item-dimension correlations were also found, with *r* values varying from 0.61 to 0.78, which indicates a moderate correlation between dimensions and their respective items. The convergent validity was examined by computing Pearson’s correlations among the GOHAI score and DMFT, DT, MT, FT, GI, PI. Discriminant validity coefficients, which are typically smaller than those of convergent validity, indicate correlations between scores of different traits.

The principal-component factor method showed that four main factors had an eigenvalue higher than 1 (respectively 3.71, 1.36, 1.34, and 1.01). 

## 4. Discussion

This study focused on the OHRQoL in elderly long-term care residents. Considering that elderly people in the area account for around 22% of the population, that the average life expectancy has lengthened and there is growing increase in public spending for the elderly, it is pivotal to understand the elderly’s perception of oral health and its link to their nutritional and psychological status. The results of the present study provided a unique opportunity for analyzing the oral impact in this group, since to the best of our knowledge this is the first study aimed at addressing the prevalence and characteristics of oral impacts in elderly long-term care residents in an area of southern Italy.

The prevalence of oral impacts experienced during the previous three months by the study population was unexpectedly low, since less than a third of the participants had a low GOHAI score, suggestive of highly compromised OHRQoL. Indeed, the GOHAI score in the present sample of elderly long-term care residents was clearly higher than that found in non-institutionalized elderly and adult populations [28,29,30,31]. Comparison with the literature is difficult and must be interpreted cautiously since the nature and magnitude of impacts could vary among populations with different socio-demographic and cultural backgrounds or different clinical conditions [32]. Perception of oral health is also related to values and expectations throughout life. Elderly people, as well as patients with chronic conditions [18], adapt to clinical changes by lowering their expectations [33], modifying what they perceive to be normal and acceptable for a given age and specific circumstances. It is well-known that, in the institutionalized elderly patients, participation in social activities is limited, and any social relationship occurs mainly with peers with similar oral health conditions. Probably, this scenario has reduced the magnitude of the impact on OHRQoL in our sample, considering that the psychosocial function of worrying about the problems with their mouth and teeth was the less affected dimension. In contrast, younger subjects have high oral health expectations, and the importance of OHRQoL is particularly relevant. It has been demonstrated that psychological impacts of oral health, such as avoiding laughing and being teased about teeth, were more prevalent in children than in adults and the elderly [34]. As reported in a previous survey conducted in the same area, younger individuals are more vulnerable to certain burdens, such as appearance, than elderly individuals [35].

In our study, more than half of the sample had poor oral health, and a significant correlation between GOHAI score and DMFT index was found, indicating that a poor oral health, as shown by a high DMFT, was correlated with a high oral impact on quality of life, as shown by a low GOHAI score. Previous surveys indicate that, among the elderly, those living in residential homes have the worst oral health conditions [36,37,38,39,40]. Poor oral health was possibly the result of inadequate connection to regular dental assistance during the stay in the facility, which has already been reported in the institutionalized elderly [41,42]. It is noticeable that poor oral health status is not gained in elderly, but it is a result of burden of oral conditions along the course of life. Moreover, oral **health** care is not universal in Italy, as well as in other countries, and it is usually quite expensive to afford. Therefore, it is urgent to prevent people lose their permanent teeth during their course of life, as well as to implement strategies to keep a good oral health status. Oral hygiene practices are substandard in older people, especially if institutionalized. Moreover, in the present study, more than half of the participants had moderate or severe cognitive impairment, and it could be argued they had a decreased ability to engage in self-care and an increased need of extra assistance with oral health care [43,44]. Indeed, only 18.4% of the participants reported frequent tooth-brushing. Improved oral health care for long-term care residents is thus urgent, and facilities should organize processes and policies to improve care providers’ knowledge and attitudes regarding oral health [45]. In a study investigating the associations of frailty with oral health, cleaning habits, and level of hygiene among home care clients aged 75 or over in Finland, those classified as frail had the benefit of preventive oral health intervention, and a positive change was observed in the prevalence of twice-daily toothbrushing and denture cleaning and especially in denture hygiene [46]. Almost one third of the participants were edentulous, and all but one were suffering from one or more missing teeth due to caries; respectively, 26.1% in the former and 42.2% in the latter group did not wear any type of prosthesis. The presence of more than 20 natural teeth has a fundamental role in maintaining a satisfactory nutritional status [47]. Indeed, masticatory performance in subjects with less than 20 teeth is lower than in those with more than 20 [48]. Missing teeth could lead to altered dietary intake and a poor nutritional status, which could contribute to increased risk of developing chronic diseases. In addition to the maintenance of the number of teeth, attempting to maintain or increase oral function as well as having a good diet and nutritional status are all linked to general health [49]. It has been observed that nutrition among these individuals may be compromised, since tooth loss affects the skill to chew effectively. In this study, physical functions such as eating certain food represents the most affected dimension of the GOHAI index, and this finding is in agreement with previous studies in elderly people [50,51] in which the GOHAI score was higher especially for physical functions. More than 40% wore dentures that also require meticulous care to mitigate the high risks of failure, inflammation, and even bone loss [52], and have to be regularly checked by a dental professional to make sure they fit and function properly.

In the present study, no association between socio-demographic factors and OHRQoL was found, although it has been demonstrated that social determinants have a strong impact on oral health [53], and oral health inequalities exist among and within different population groups [54] and through the entire life course [55]. Further research is needed to evaluate the potential role of socio-demographic factors on the oral health status and on OHRQoL in the institutionalized geriatric patients.

### Strengths and Limitations

The strengths of the present study are the size of the study population, its social diversity, and the response rate. The high response rate is extremely satisfactory and restricts one major potential source of bias in the results. It remains an important indicator of survey quality, and we believe that time and effort spent by survey researchers to improve it has made this possible. One limitation is in the cross-sectional design, and thus we have to be careful about interpreting the associations and direction of associations from the survey. Second, we found that the vast majority of participants were in the >80 years age group and we acknowledge that other age groups have been underrepresented. For future studies, the authors recommend the sample size to be larger to include enough participants stratified by different socioeconomic background and age groups. Third, this was an epidemiological survey and the oral examinations were carried out in the long-term care facility with basic equipment and under field conditions, rather than in a dental clinic with the use of extensive diagnostic tools. This may have caused underestimation of the extent of oral diseases. Similarly, the self-assessed need for dental treatment and OHRQoL by elderly subjects who had moderate or severe cognitive impairment may have introduced an underestimation of the impact of oral problems. However, GOHAI has already been successfully used to evaluate OHRQoL among the elderly diagnosed with dementia [29] and Alzheimer’s disease [56]. Moreover, further attention is needed to improve the oral health status of the elderly with dementia, since higher level of plaque, coronal and root caries, retained gingival, and periodontal disease are highly common in these patients [57]. Fourth, the impact of edentulism on daily oral function and social interactions has significant plausibility and the relevant proportion of edentulous individuals that did not wear any type of prosthesis could have affected the study results. However, research on this topic is limited and further evidence is strongly needed. Finally, the data were collected in one Italian region and concern about generalizability of our results may arise. Therefore, although we cannot exclude that our results pertain only to our area, it is reasonable to suppose that an analogous context may be referred to the southern part of our country. To have more insight into the impact of oral health on the OHRQoL, we strongly suggest a replication of the study in other regions of the country.

## 5. Conclusions

Despite these limitations, the results of this study have some implications for public health professionals, gerontologists, and dental practitioners. They documented poor oral health, with a high prevalence of missing teeth and denture-wearing, gingival inflammation, and an unacceptable level of oral hygiene and denture care among long-term care facility residents, together with a lack of perceived impact on their OHRQoL. Since front-line care providers are responsible for the majority of hands-on daily care, including oral hygiene, they can play a pivotal role in improving oral health care for these elderly individuals. Evidence-based interventions and strategies to improve oral health should not only target the long-term care facility residents, but also the health professionals and caregivers responsible for these individuals. Moreover, dentists’ and dental hygienists’ regular visits in long-term care facilities could improve oral health throughout a multidisciplinary approach to oral care.

## Figures and Tables

**Table 1 ijerph-18-02175-t001:** Frequency distribution of sociodemographic and clinical variables according to the Geriatric Oral Health Assessment Index (GOHAI) score.

Variables	Total	Good GOHAI Score (>50)	Poor GOHAI Score (≤50)	*p* Value
	*N* (%)	*N* (%)	*N* (%)	
**Age (years)** <80 ≥80	85 (29.5) 203 (70.5)	59 (69.4) 145 (71.4)	26 (30.6) 58 (28.6)	0.731
**Gender** Female Male	195 (67.7) 93 (32.3)	139 (71.3) 65 (69.9)	56 (28.7) 28 (30.1)	0.808
**Marital status** Widower Other	149 (51.7) 139 (48.3)	108 (72.5) 96 (69)	41 (27.5) 43 (31)	0.524
**Education level** None Primary/Middle school High school/University Degree	111 (38.5) 134 (35.8) 43 (25.7)	77 (69.4) 94 (70.1) 33 (76.7)	34 (30.6) 40 (29.9) 10 (23.3)	0.646
**Previous occupation** Employed Unemployed/Housewife	213 (74) 75 (26)	152 (71.4) 52 (69.3)	61 (28.6) 23 (30.7)	0.740
**Smoking habit** Never smoker Past or Current smoker	204 (71) 84 (29)	139 (68.1) 65 (77.4)	65 (31.9) 19 (22.6)	0.117
**Cognitive impairment** None Mild Moderate Severe	52 (19.8) 74 (28) 121 (45.8) 17 (6.4)	38 (73.1) 53 (71.6) 86 (71.1) 10 (58.8)	14 (26.9) 21 (28.4) 35 (28.9) 7 (41.2)	0.720
**Frequency of tooth brushing** Less than once a day Once a day More than once a day	71 (35.7) 75 (37.7) 53 (26.6)	46 (64.8) 52 (69.3) 39 (73.6)	25 (35.2) 23 (30.7) 14 (26.4)	0.751
**Self reported need of dental care** Yes No	115 (39.9) 173 (60.1)	59 (51.3) 145 (83.8)	56 (48.7) 28 (16.2)	<0.001
**Use of prosthesis**				0.025
No prosthesis	167 (58)	107 (64.1)	60 (35.9)	
Removable Partial denture Removable Complete denture	20 (20.4) 78 (79.6)	16 (80) 64 (82.1)	4 (20) 14 (17.9)	
Fixed Partial denture	23 (8)	10 (83.9)	6 (26.1)	
**Oral status**				0.001
Edentulous Wearing denture Not wearing denture	92 (31.9) 68 (73.9) 24 (26.1)	57 (83.8) 12 (50)	11 (16.2) 12 (50)	

**Table 2 ijerph-18-02175-t002:** Distribution of GOHAI by domains.

GOHAI Score ≤50 >50	*N* (%) 84 (29.2) 204 (70.8)		Mean ± SD 53.3 ± 6.3
Items *	Never and Rarely (%)	Sometimes, Often and Always (%)	Mean ± SD
*Physical function*			
How often did you limit the kinds or amounts of food you eat because of problems with your teeth or dentures?	148 (54.9)	130 (45.1)	3.7 ± 1.3
How often did you have trouble biting or chewing any kinds of food, such as firm meat or apples?	147 (51)	141 (49)	3.6 ± 1.3
How often were you able to swallow comfortably? ^#^	4 (1.4)	284 (98.6)	4.7 ± 0.7
How often have your teeth or dentures prevented you from speaking the way you wanted?	248 (86.1)	40 (13.9)	4.6 ± 0.9
*Pain or discomfort*			
How often were you able to eat anything without feeling discomfort? ^#^	19 (6.6)	269 (93.4)	4.4 ± 1
How often did you use medication to relieve pain or discomfort from around your mouth?	273 (94.8)	15 (5.2)	4.8 ± 0.5
How often were your teeth or gums sensitive to hot, cold, or sweets?	245 (85.1)	43 (14.9)	4.6 ± 0.9
*Psychosocial function*			
How often did you limit contacts with people because of the condition of your teeth or dentures?	276 (95.9)	12 (4.1)	4.8 ± 0.5
How often were you pleased or happy with the looks of your teeth and gums, or dentures? ^#^	31 (10.8)	257 (89.2)	4.4 ± 1.1
How often were you worried or concerned about the problems with your teeth and gums, or dentures?	195 (67.7)	93 (32.3)	4.1 ± 1.2
How often did you feel nervous or self-conscious because of the problems with teeth and gums, or dentures?	235 (82)	52 (18)	4.5 ± 1
How often did you feel uncomfortable eating in front of people because of problems with teeth or dentures?	274 (95.2)	14 (4.8)	4.8 ± 0.5

* In the past 3 months. GOHAI = Geriatric Oral Health Assessment Index. ^#^ Inversely scaled questions.

**Table 3 ijerph-18-02175-t003:** Multiple logistic regression analysis results examining GOHAI score according to several variables.

Variable	OR	SE	95% CI	*p*
***Model: Overall impact score measured with GOHAI***				
Log likelihood = −141.38, χ^2^ = 64.9, *p* < 0.00001, No. of obs = 288				
**Gender**	Backward elimination			
**Age, years**	Backward elimination			
**Marital status**	Backward elimination			
**Education level**	Backward elimination			
**Previous occupation**	Backward elimination			
**Cognitive impairment**	Backward elimination			
**Use of prosthesis**	Backward elimination			
**PI**	Backward elimination			
**GI**	Backward elimination			
**Smoking habit**				
Never smoker	1.00			
Past or Current smoker	2.65	1.35	0.97–7.22	0.057
**Self reported need of dental care**				
No	1.00			
Yes	0.20	0.07	0.10–0.42	<0.001
**DMFT Index**	0.88	0.03	0.83–0.94	<0.001

## Data Availability

The dataset used and analyzed during the current study is available from the corresponding author upon request.

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
