# Peer review of "Oral Health Status and the Impact on Oral Health-Related Quality of Life among the Institutionalized Elderly Population: A Cross-Sectional Study in an Area of Southern Italy"

_ijerph, 2021, doi:10.3390/ijerph18042175_

Round 1
Reviewer 1 Report
The authors have described a random cluster sampling methodology. But there is no mention about the sampling power ascertained during sample size estimation.
Similarly, the authors seem to have reverse worked their sample size based on the number of survey samples with positive response. If not so, why there was no additional respondents included (10% overestimation) to improve statistical strength of the study?
Author Response
We have accordingly attached a copy of the edited version of the paper. All suggestions made by the Reviewers have been taken into account and were marked in bold.
Reviewer: 1
The authors have described a random cluster sampling methodology. But there is no mention about the sampling power ascertained during sample size estimation.
Similarly, the authors seem to have reverse worked their sample size based on the number of survey samples with positive response. If not so, why there was no additional respondents included (10% overestimation) to improve statistical strength of the study?
In response to the point regarding the cluster structure of the data, we took it into account when calculating the sample size. Indeed, the sample size calculated for simple random sampling (288 subjects) was corrected for a design effect of 1.2, yielding a sample size of 345. The magnitude of the design effect was chosen considering a low intracluster correlation, since it was expected that there were no substantial differences among the elderly populations institutionalized in the different facilities (Aday, L.A.; Cornelius, L.J. Designing and Conducting Health Surveys: A Comprehensive Guide, 3rd ed.; Jossey-Bass Publisher: San Francisco, CA, USA, 2006). These considerations have now been clarified in the Methods section (page 2, Methods, Study population and sampling).
Reviewer 2 Report
We read with great interest the study “Oral health status and the impact on oral health related quality 2 of life among the institutionalized elderly population: a cross- 3 sectional study in Italy” aiming to describe the oral health status in the 22 Italian institutionalized geriatric population and to identify the impact of oral health on the Oral 23 Health Related Quality of Life (OHRQoL).
Overall the study is well written and well conducted, however some critical issues require the authors to amend.
First of all, avoid using colloquial forms such as "we" in the text.
In the abstract, the age range is defined as> 60 years, but the sample, as described in the results, is greater than 70 years. Please change this in the abstract and in the text.
In addition, the research in the title claims to represent Italy, but was carried out in the southern part, therefore, taking into account the differences between the Italian regions and taking into account that out of the 20 regions only one was surveyed, please amend this detail in the title, the abstract and the text, clarifying that the survey carried out is representative of southern Italy.
In the discussion it is said that the older part of the population constitutes 20% of the population, it is asked to verify this data for the region of Calabria in which the study was conducted.
Moreover, it is necessary to add to the limitations of the study the fact that the study itself cannot indicate the southern Italy as it only concerns the elderly population living in the Calabria region.
Author Response
We have accordingly attached a copy of the edited version of the paper. All suggestions made by the Reviewers have been taken into account and were marked in bold.
Reviewer: 2
First of all, avoid using colloquial forms such as "we" in the text.
As suggested, we avoided the “we” colloquial forms throughout the manuscript.
In the abstract, the age range is defined as> 60 years, but the sample, as described in the results, is greater than 70 years. Please change this in the abstract and in the text.
It is true that the majority of the sample was in the ≥ 80 years age group (70.5%), but we clarified that the age range is 61- 99 years (page 5, 1st paragraph of results)
In addition, the research in the title claims to represent Italy, but was carried out in the southern part, therefore, taking into account the differences between the Italian regions and taking into account that out of the 20 regions only one was surveyed, please amend this detail in the title, the abstract and the text, clarifying that the survey carried out is representative of southern Italy.
As suggested, we clarified that the study pertains to an area of Southern Italy, in the title, the abstract, in the introduction (page 2, last paragraph) and the discussion (page 8, 1st paragraph).
In the discussion it is said that the older part of the population constitutes 20% of the population, it is asked to verify this data for the region of Calabria in which the study was conducted.
As suggested, we highlighted that the elderly individuals represent around 22% of the regional population (page 7, last paragraph).
Moreover, it is necessary to add to the limitations of the study the fact that the study itself cannot indicate the southern Italy as it only concerns the elderly population living in the Calabria region.
As suggested, in the limitation section, we highlighted that the data were collected in one Italian region and concern about generalizability of our results may arise. Therefore, although we cannot exclude that our results pertain only to our area, it is reasonable to suppose that an analogous context may be referred to the Southern part of our country. To have more insight into the impact of oral health on the OHRQoL, we strongly suggest replication of the study in other regions of the country. (page 9, last paragraph).
Reviewer 3 Report
This is an extremely relevant subject with an impressive sample; it deserves publication and requires only minor adjustments. Below are suggestions to improve the article. Congratulations on your beautiful study!
- Was it a single interviewer? If so, what was the interviewer's training? How was the interviewer calibrated?
- Were oral lesions, medication and dental floss use also investigated?
- What is the length of stay for the elderly? Could this factor influence the results?
- Fifty-two percent have moderate to severe cognitive illness. Can this data impact the result? Would these patients be able to perform adequate oral hygiene? As well as assessing quality of life?
- Limiting points are the high prevalence of moderate to severe cognitive illness, high rate of edentulous and non-prosthetic users.
- The lack of a control group does not allow us to extrapolate the results, since we have a local sample, predominantly of women with a socio-economic profile.
- Insert researches from the year 2020
Author Response
We have accordingly attached a copy of the edited version of the paper. All suggestions made by the Reviewers have been taken into account and were marked in bold.
Reviewer: 3
This is an extremely relevant subject with an impressive sample; it deserves publication and requires only minor adjustments. Below are suggestions to improve the article. Congratulations on your beautiful study!
We thank the reviewer for the positive tone of the comments.
Was it a single interviewer? If so, what was the interviewer's training? How was the interviewer calibrated?
In response to this point, we clarified that questionnaire was administered by four trained and calibrated interviewers, and we highlighted that the training involved a theoretical phase (presentation and explanation of the instrument) and a practical phase (practice with other interviewers, shadow interview, reverse shadow). To assure the calibration, each interviewer repeated 10 interviews after one week in order to analyze the intra- and inter-rater agreement. (pages 2 and 3)
We used the kappa coefficient to calculate the intra- and inter-rater agreement. (page 5, 1st paragraph).
The intra- and inter-rater agreement rates varied from 0.891 to 0.948. (page 6, 1st paragraph).
Were oral lesions, medication and dental floss use also investigated?
Within the pain or discomfort domain of the GOHAI, we investigated how often elderly individuals have used medication to relieve pain or discomfort from around your mouth (Table 2).
What is the length of stay for the elderly? Could this factor influence the results?
In response to this point, we clarified the long-term facility is a setting where people depend on help with daily living activities and/or are in need of some permanent nursing care. Especially individuals over 75 years of age are more likely to develop chronic pathologies, comorbidities or other impairing diseases, that require continuous assistance until death (page 2, 2nd paragraph).
Therefore, we did not explore the length of stay of the participants.
Fifty-two percent have moderate to severe cognitive illness. Can this data impact the result? Would these patients be able to perform adequate oral hygiene? As well as assessing quality of life?
We thank the reviewer for the comment. We pointed out in the method section that “Elderly subjects who had moderate or severe cognitive impairment were assisted by their caregivers in answering the questions about oral hygiene habits and GOHAI items, for preventing information bias.” (page 3, 1st paragraph). Moreover, in the discussion section, we clarified that more than half of the participants had moderate or severe cognitive impairment, and it could be argued they had a decreased ability to engage in self-care and an increased need of extra assistance with oral health care. Indeed, only 18.4% of the participants reported frequent tooth-brushing. (page 8, last paragraph). In addition, in the study limitations we highlighted that self-assessed OHRQoL by elderly subjects who had moderate or severe cognitive impairment may have introduced underestimation of impact of oral problems. (page 9)
Limiting points are the high prevalence of moderate to severe cognitive illness, high rate of edentulous and non-prosthetic users.
We agree with the reviewer and in the study limitations we pointed out that self-assessed need for dental treatment and OHRQoL by elderly subjects who had moderate or severe cognitive impairment may have introduced underestimation of impact of oral problems, as above mentioned. However, GOHAI has already been successfully used to evaluate OHRQoL among elderly diagnosed with dementia and Alzheimer's disease. Moreover, further attention is needed to improve the oral health status of the elderly with dementia, since higher level of plaque, coronal and root caries, retained gingival, and periodontal disease are very common in these patients (page 9, limitations section).
Regarding edentulism, we highlighted that the impact of edentulism on daily oral function and social interactions has significant plausibility and the relevant proportion of edentulous individuals that did not wear any type of prosthesis could have affected the study results. However, research on this topic is limited and further evidence is strongly needed. (page 9, last paragraph).
The lack of a control group does not allow us to extrapolate the results, since we have a local sample, predominantly of women with a socio-economic profile.
In response to this point, we have clarified that the data were collected in one Italian region and concern about generalizability of our results may arise. Therefore, although we cannot exclude that our results pertain only to our area, it is reasonable to suppose that an analogous context may be referred to the Southern part of our country. To have more insight into the impact of oral health on the OHRQoL, we strongly suggest replication of the study in other regions of the country. (page 9, last paragraph).
Insert researches from the year 2020
As suggested, we cited and commented the latest following research about the topic:
- Tuuliainen, E.; Nihtilä, A.; Komulainen, K.; Nykänen, I.; Hartikainen, S.; Tiihonen, M.; Suominen, A.L. The association of frailty with oral cleaning habits and oral hygiene among elderly home care clients. Scand J Caring Sci 2020, 34, 938–947.
- Aquilanti, L.; Alia, S.; Pugnaloni, S.; Coccia, E.; Mascitti, M.; Santarelli, A.; Limongelli, L.; Favia, G.; Mancini, M.; Vignini, A.; Rappelli, G. Impact of Elderly Masticatory Performance on Nutritional Status: An Observational Study. Medicina 2020, 56, 1–9.
- Watanabe, Y.; Okada, K.; Kondo, M.; Matsushita, T.; Nakazawa, S.; Yamazaki, Y. Oral health for achieving longevity Gerontol. Int. 2020, 20, 526–538.
Round 2
Reviewer 2 Report
The text has been amended. The article may be accepted in the present form.